# Exploring the Spatiotemporal Dynamics of CO$_2$ Emissions through a Combination of Nighttime Light and MODIS NDVI Data

Yongxing Li [1], Wei Guo [1,2,*], Peixian Li [1], Xuesheng Zhao [1] and Jinke Liu [1]

[1] College of Geoscience and Surveying Engineering, China University of Mining & Technology, Beijing 100083, China
[2] Chinese Academy of Surveying & Mapping, Beijing 100830, China
* Correspondence: weiguo@cumtb.edu.cn

**Abstract:** Climate change caused by CO$_2$ emissions is posing a huge challenge to human survival, and it is crucial to precisely understand the spatial and temporal patterns and driving forces of CO$_2$ emissions in real time. However, the available CO$_2$ emission data are usually converted from fossil fuel combustion, which cannot capture spatial differences. Nighttime light (NTL) data can reveal human activities in detail and constitute the shortage of statistical data. Although NTL can be used as an indirect representation of CO$_2$ emissions, NTL data have limited utility. Therefore, it is necessary to develop a model that can capture spatiotemporal variations in CO$_2$ emissions at a fine scale. In this paper, we used the nighttime light and the Moderate Resolution Imaging Spectroradiometer (MODIS) normalized difference vegetation index (NDVI), and proposed a normalized urban index based on combination variables (NUI-CV) to improve estimated CO$_2$ emissions. Based on this index, we used the Theil–Sen and Mann–Kendall trend analysis, standard deviational ellipse, and a spatial economics model to explore the spatial and temporal dynamics and influencing factors of CO$_2$ emissions over the period of 2000–2020. The experimental results indicate the following: (1) NUI-CV is more suitable than NTL for estimating the CO$_2$ emissions with a 6% increase in average $R^2$. (2) The center of China's CO$_2$ emissions lies in the eastern regions and is gradually moving west. (3) Changes in industrial structure can strongly influence changes in CO$_2$ emissions, the tertiary sector playing an important role in carbon reduction.

**Keywords:** CO$_2$ emissions; normalized urban index based on combination variables; standard deviational ellipse; Theil–Sen and Mann–Kendall trend analysis; nighttime light

## 1. Introduction

Global warming, glacial melting, and ocean level rise caused by climate change have created great threats to human survival and development [1–3]. If global temperatures continue to rise, it is possible that more extreme weather events will occur. How to deal with climate change has become a thorny issue for scholars from all over the world [4,5]. It has been widely accepted that CO$_2$ emissions are a major cause of climate change and that fossil fuel combustion is the main contributor to CO$_2$ emissions [6,7]. Curbing energy-related CO$_2$ emissions is a necessary path to achieve carbon neutrality and is the key to mitigating the human climate crisis [8]. As its economy grows, China's demand for fossil fuels is also increasing. At present, China has the highest level of CO$_2$ emissions in the world [9–11]. In the context of climate change, China has formulated many measures to reduce CO$_2$ emissions. In 2015, the Chinese government showed its determination to save energy and reduce emissions. It pledged to achieve a 60–65% reduction in CO$_2$ emissions intensity by 2030 compared to 2005 [12]. Therefore, an accurate and comprehensive understanding of the spatiotemporal distribution and driving forces of CO$_2$ emissions is a prerequisite for implementing precise emission reduction strategies [13–15].

In different parts of a city, $CO_2$ emissions may have different spatial dynamics, but statistical data can only show the overall development of a city and cannot reveal the urban internal landscapes [16]. Therefore, how to investigate $CO_2$ emissions at a fine-scale has become an important topic of concern to many scholars. It has been pointed out that nighttime light data (e.g., the Defense Meteorological Satellite Program's Operational Linescan System [DMSP-OLS] and the Visible Infrared Imaging Radiometer Suite Day/Night Band [VIIRS-DNB]) can describe socio-economic phenomena related to human activities [17]. These data are widely used in studies on urbanization [18–20], poverty analysis [21,22], impervious surface extraction [23–25], environmental variations [26–28], population spatialization [29,30], electric power consumption [31,32] and economic analysis [33–35]. Since nighttime light is highly correlated with human activity, and $CO_2$ emissions are an inevitable product of human activities, a connection can be made between nighttime light with $CO_2$ emissions [36,37]. Doll et al. found a high correlation between $CO_2$ emissions and DMSP-OLS nighttime light data, and mapped the distribution of $CO_2$ emissions at a global scale [38]. Ghosh et al. introduced population distribution data to model $CO_2$ emissions from people in areas with more or less nighttime light [39]. However, their method was relatively simple and the DMSP-OLS data have limitations in estimating $CO_2$ emissions [40,41]. The spatial resolution of DMSP-OLS is about 1000 m, and it cannot distinguish pixels with values larger than 63. All values larger than 63 are recorded as 63, and the phenomenon is the saturation effect [42,43]. Due to the saturation effect and coarse spatial resolution of DMSP-OLS, it fails to reflect the differences in core urban areas in detail [32,44]. Although many scholars have proposed various correction methods for this problem, these methods may provide quite different results [45–47]. VIIRS-DNB not only has a higher spatial resolution, but also has a longer detection range without saturation effects [48]. Since the release of VIIRS-DNB data, the higher spatial resolution and longer detection range have motivated scholars to perform higher resolution studies [49,50]. Shi et al. demonstrated that VIIRS-DNB is a powerful indicator for modeling socioeconomic phenomena [51].

Although nighttime light data can study the $CO_2$ emissions in cities, there are still some shortcomings in using nighttime light data alone [12,24,52,53]. More and more scholars are introducing vegetation cover data (e.g., normalized difference vegetation index [NDVI]) into their studies [23,54]. Experiments have shown that this combination provides good results in impervious surface area (ISA) mapping. However, few surveys use this method to estimate $CO_2$ emissions. Based on the existing studies, some scholars have studied the changes in $CO_2$ emissions at different scales, but they only summarized $CO_2$ emissions of different administrative units, and the trend of $CO_2$ emissions on pixels is still unclear [7,16,55]. Although some scholars have simply analyzed the change of $CO_2$ emissions, significant analysis of trends in changes is still lacking [10,47]. Additionally, the analysis of the spatial and temporal dynamic patterns and the influencing factors related to $CO_2$ emissions is not comprehensive [56]. Therefore, it is necessary to conduct an in-depth and systematic analysis of the influencing factors related to $CO_2$ emissions [57–59].

In this study, a new combination index was proposed to improve the estimation accuracy of $CO_2$ emissions. We then aimed to explore the trends and significance in $CO_2$ emissions at the pixel scale. Furthermore, we investigated the spatial and temporal dynamics of $CO_2$ emissions and the drivers of $CO_2$ emissions were discussed.

## 2. Study Area and Datasets

Mainland China is the main area under examination in this paper and we divided the study area into three major regions based on the research by Guo et al. [53]. The detailed region division results are shown in Figure 1. The datasets used in this study are nighttime light imageries, MODIS NDVI product, energy consumption statistics, socioeconomic data and administrative boundaries.

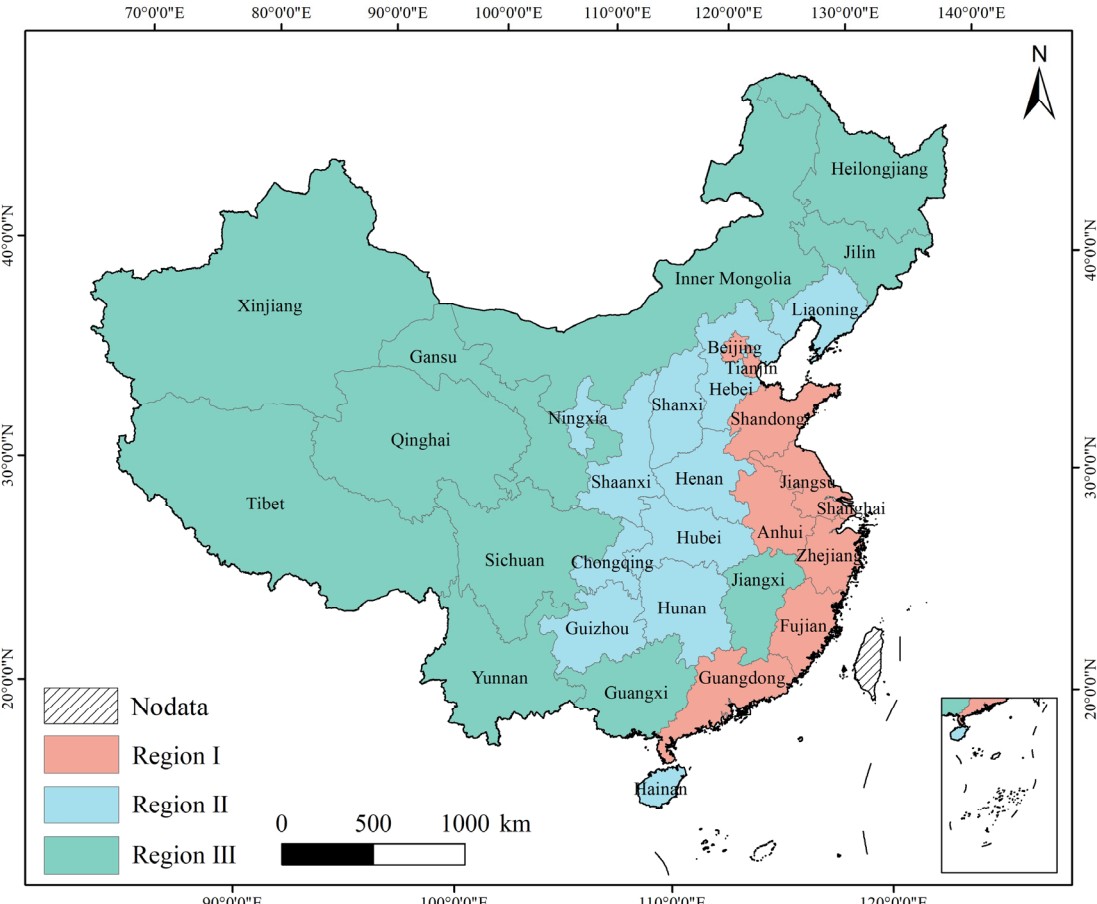

**Figure 1.** Study area.

Nighttime light data from 2000 to 2020 were used, referring to the method proposed by Chen et al. [60], and the original dataset of 2000–2012 was taken from the enhanced vegetation index-adjusted NTL index (EANTLI) dataset [61]. By integrating DMSP-OLS and enhanced vegetation index (EVI) data, EANTLI can reduce the saturation effect of DMSP-OLS and enhance image information:

$$EANTLI = \frac{1 + (NTL_{nor} - EVI)}{1 - (NTL_{nor} - EVI)} \times NTL \qquad (1)$$

where $NTL_{nor}$ denotes the normalized DMSP-OLS, $EVI$ is an annual data of EVI, and $NTL$ is the original nighttime light brightness.

Monthly VIIRS-DNB data after 2013 were used to produce an annual median VIIRS-DNB dataset to obtain more reliable annual data. Then, the annual VIIRS-DNB products were treated with reference to Shi et al. [51], who proposed an auto-encoder model with convolutional neural networks (CNN) to transform EANTLI data into VIIRS-DNB-like data. The data from 2013 and 2012 were used as the training and testing sets, respectively. This dataset can describe the dynamic changes of socio-economic characteristics accurately over a long time period with a resolution of 15 arc-second (~500 m). Although NTL can be used as an indirect representation of $CO_2$ emissions, NTL data have limited utility. Therefore, it is necessary to introduce additional data to capture the spatiotemporal variations in $CO_2$ emissions at a fine scale. Among them, vegetation cover data is regarded as a data source that can effectively supplement the nighttime light information.

MODIS NDVI (MOD13A1) data were downloaded from the Google Earth Engine platform and produced as an annual product. The energy consumptions were obtained from the China Energy Statistics Yearbook, which describes eight energy types and can

be used to estimate $CO_2$ emissions. Statistics, such as year-end population, added value of the secondary industry (AVSI), and added value of the tertiary industry (AVTI) were downloaded from the China National Bureau of Statistics. The administrative boundary data of the country and its provinces were taken from China's National Geomatics Centre. The details of the dataset are summarized in Table 1.

**Table 1.** Datasets used in research.

| Data | Description | Year | Source |
|---|---|---|---|
| Nighttime light (DMSP-OLS, VIIRS-DNB) | Long time series of global nighttime light data. | 2000–2020 | https://dataverse.harvard.edu/dataset.xhtml?persistentId=doi:10.7910/DVN/YGIVCD (accessed on 13 August 2023) |
| MODIS NDVI (MOD13A1) | Global 500 m spatial resolution 16-day product. | 2000–2020 | Google Earth Engine platform (https://code.earthengine.google.com/, accessed on 13 August 2023) |
| Energy consumption data | Energy statistics for 30 provinces ($10^4$ t). | 2000–2020 | China Energy Statistics Yearbook |
| Socioeconomic data | Three types of socio-economic indicators: population, AVSI, and AVTI. | 2000, 2010, 2020 | China National Bureau of Statistics |

## 3. Methods

There were four steps in our research methods. First, we preprocessed the remote sensing data to maintain the same resolution and coordinate system. Second, $CO_2$ emissions were calculated from energy consumption. Third, we investigated the spatiotemporal dynamics of $CO_2$ emissions. Finally, the driving forces behind $CO_2$ emissions were discussed. The detailed process is shown in Figure 2.

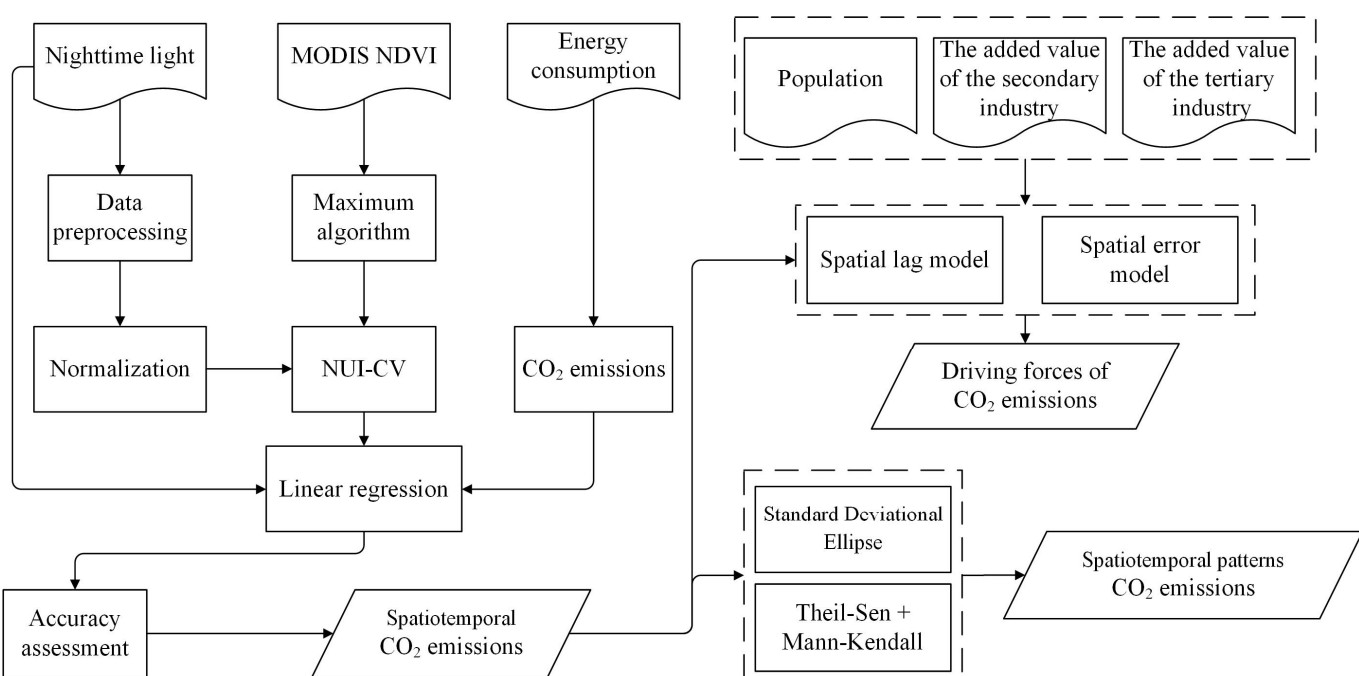

**Figure 2.** Methodological framework.

### 3.1. Preprocessing of Remote Sensing Data

For the uniformity of projection, all remote sensing data (i.e., nighttime light and MODIS NDVI) were reprojected to the Albers Conical Equal Area projection and resampled to 500 m resolution.

In order to avoid the impacts of outliers, we set the maximum of nighttime light to 100, and all pixels with DN values greater than 100 were set to 100. The reason for setting this threshold is that most pixel values in the nighttime light imageries are less than 100 [24]. After that, the nighttime light data were normalized with Equation (2):

$$NTL_{nor} = \frac{NTL - NTL_{min}}{NTL_{max} - NTL_{min}} \tag{2}$$

where $NTL_{nor}$ is the normalized nighttime light data, with a range of 0–1; and $NTL_{max}$ and $NTL_{min}$ represent the maximum and minimum pixel values, respectively.

To avoid the confusion of bare soil, water bodies, and ISA, we used the maximum value method to composite the multi-period NDVI images:

$$NDVI_{max} = MAX[NDVI_1, NDVI_2, \cdots, NDVI_n] \tag{3}$$

where $NDVI_1$, $NDVI_2$, …, $NDVI_n$ are the multitemporal MOD13A1 NDVI images.

*3.2. Estimation of CO$_2$ Emissions*

We used the formula developed by the Intergovernmental Panel on Climate Change (IPCC) to compute statistical CO$_2$ emissions [62]. The statistical CO$_2$ emissions (*SC*) can be formulated by:

$$SC = \sum_{w=1}^{W} E_w \times CEC_w \times ALC_w \tag{4}$$

where *E* represents the amount of energy consumption, $\omega$ denotes the energy types, and *CEC* and *ALC* are the carbon emission coefficients and the average low-order calorific values, respectively.

A new index is presented here, named the Normalized Urban Index Based on Combination Variables (NUI-CV), which combines the nighttime light (NTL) and MODIS NDVI datasets:

$$\text{NUI-CV} = (1 - NDVI_{max}) \times log_2\left(1 + \sqrt{NTL_{nor}}\right) \tag{5}$$

Since $NDVI_{max}$ is negatively associated with urban sprawl, $1 - NDVI_{max}$ (ranging between 0 and 1) is positively associated with urban information. $log_2\left(1 + \sqrt{NTL_{nor}}\right)$ not only smooths the extreme values of nighttime light, but also holds the value between 0–1. The information on human activities can be enhanced by integrating NTL and NDVI datasets in this way. To verify the accuracy of NUI-CV, we modeled NTL and NUI-CV, respectively, with CO$_2$ emissions using a linear regression model. It is because linear regression models are able to describe the relationship between different variables intuitively and are simple to implement and widely used.

*3.3. Assessment of Spatiotemporal Dynamics of CO$_2$ Emissions*

3.3.1. Analysis of CO$_2$ Emissions Trend

The Theil–Sen and Mann–Kendall (TS-MK) trend analysis method included Theil–Sen slope estimation and the Mann–Kendall test [63]. Theil–Sen slope estimation is generally employed to calculate the trend value, and it is insensitive to outliers in the series dataset. However, it cannot provide significance judgments and usually needs to be performed together with the Mann–Kendall test [64]. The Mann–Kendall test is a non-parametric time series trend test that provides the significance of the trend of change. TS-MK is often used in the studies related to vegetation cover and climate change. In this paper, the method is used to detect the significance of the change trend of CO$_2$ emissions.

The Theil–Sen slope formula is:

$$\beta = Median\left(\frac{x_j - x_i}{j - i}\right), j > i \tag{6}$$

where $\beta$ represents the trend degree; when $\beta$ is greater than 0 means that $CO_2$ emissions are increasing over time, when $\beta$ is less than 0 means that $CO_2$ emissions are decreasing; $x_i$ and $x_j$ represent the carbon emissions in years $i$ and $j$.

The statistical values associated with Mann–Kendall test are calculated, as shown in equation.

$$S = \sum_{i=1}^{n-1} \sum_{j=i+1}^{n} sgn(x_j - x_i), sgn(x_j - x_i) = \begin{cases} +1, & x_j > x_i \\ 0, & x_j = x_i \\ -1, & x_j < x_i \end{cases} \tag{7}$$

$$Z = \begin{cases} \frac{S-1}{\sqrt{VAR(S)}}, & S > 0 \\ 0, & S = 0, Var(S) = \frac{n(n-1)(2n+5)}{18} \\ \frac{S-1}{\sqrt{VAR(S)}}, & S > 0 \end{cases} \tag{8}$$

where $n$ is the time period (2000–2020), $S$ is the test statistic, $Var(S)$ is the variance of $S$, and $Z$ denotes the significance.

According to different confidence levels, $\beta$ and $Z$ are classified into different categories. The detailed categories are found in Table 2.

**Table 2.** Categories of changes in $CO_2$ emission trends.

| $\beta$ | $Z$ | Trend Category |
|---|---|---|
| $\beta > 0$ | $2.58 < |Z|$ | Extremely significant increase |
| | $1.96 < |Z| \leq 2.58$ | Significant increase |
| | $1.65 < |Z| \leq 1.96$ | Slightly significant increase |
| | $|Z| \leq 1.96$ | Not significantly increased |
| $\beta = 0$ | Any value | No change |
| $\beta < 0$ | $|Z| \leq 1.96$ | Not significantly decrease |
| | $1.65 < |Z| \leq 1.96$ | Slightly significant decrease |
| | $1.96 < |Z| \leq 2.58$ | Significant decrease |
| | $2.58 < |Z|$ | Extremely significant decrease |

### 3.3.2. $CO_2$ Emissions Evolution Direction

Standard deviational ellipse (SDE) can describe the spatial distribution of data from multiple directions [65]. By analyzing various parameters related to SDEs, the direction and the change in the distribution trend of $CO_2$ emissions can be obtained [66]. The weighted mean center is the center of the spatial distribution and can be calculated using Equation (6):

$$M(\hat{x}, \hat{y}) = \left( \frac{\sum_{i=1}^{n} w_i x_i}{\sum_{i=1}^{n} w_i}, \frac{\sum_{i=1}^{n} w_i y_i}{\sum_{i=1}^{n} w_i} \right) \tag{9}$$

where $M(\hat{x}, \hat{y})$ is the weighted mean center, $x_i$ and $y_i$ are the coordinates of spatial unit $i$, $\omega$ denotes the spatial weight, and $n$ represents the sum of spatial units. The rotation angle can be described with $\tan\theta$ as follows:

$$\tan\theta = \frac{\left( \sum_{i=1}^{n} \widetilde{x}_i^2 - \sum_{i=1}^{n} \widetilde{y}_i^2 \right) + \sqrt{\left( \sum_{i=1}^{n} \widetilde{x}_i^2 - \sum_{i=1}^{n} \widetilde{y}_i^2 \right)^2 + 4\left( \sum_{i=1}^{n} \widetilde{x}_i \widetilde{y}_i \right)^2}}{2\sum_{i=1}^{n} \widetilde{x}_i \widetilde{y}_i} \tag{10}$$

where $\theta$ is the azimuth angle of the ellipse, and $\widetilde{x}_i$ and $\widetilde{y}_i$ denote the deviation of the XY coordinate from the weighted mean center.

$$\delta_x = \sqrt{\frac{\sum_{i=1}^{n}\left(w_i\widetilde{x}_i\cos\theta - w_i\widetilde{y}_i\sin\theta\right)^2}{\sum_{i=1}^{n} w_i^2}} \tag{11}$$

$$\delta_y = \sqrt{\frac{\sum_{i=1}^{n}\left(w_i\widetilde{x}_i\sin\theta - w_i\widetilde{y}_i\cos\theta\right)^2}{\sum_{i=1}^{n} w_i^2}} \tag{12}$$

In the above equation, $\delta_x$ and $\delta_y$ are the standard deviations of the ellipse x-axis and y-axis, respectively.

### 3.4. Driving Force Analysis of $CO_2$ Emissions

In contrast to the traditional econometrics, spatial econometric models are able to adequately account for spatial dependence [67]. The first law of geography proposed by Tobler et al. highlights the existence of interactions between spatial units and provides a theoretical basis for spatial measurement [68]. To investigate how the spatial effects influence $CO_2$ emissions, here the spatial lag model (SLM) and spatial error model (SEM) were introduced into the experiment. The SLM represents the impact of $CO_2$ emissions from surrounding provinces on $CO_2$ emissions in a particular city [69]. The model is specified as follows:

$$Y = \rho WY + X\beta + \mu \tag{13}$$

where $X$ and $Y$ represent the dependent and independent variable matrices, respectively, $\rho$ denotes the spatial effect coefficient, $W$ represents the spatial matrix, $\beta$ is the parameter vector, and $\mu$ denotes the random error vector, satisfying $\mu \sim N\left(0, \sigma^2\right)$.

The SEM assumes that the spatial error term is correlated with the spatial totality [70]. The error of an individual will affect other individuals with spatial effects:

$$Y = X\beta + \varepsilon \tag{14}$$

$$\varepsilon = \lambda W\varepsilon + \mu \tag{15}$$

where $\lambda$ represents the coefficient of spatial error term and $\varepsilon$ is the error term of spatial auto-correction.

Previous studies have indicated that differences in population and industrial structure are major contributors to $CO_2$ emissions [12,53]. However, few scholars constructed SLM and SEM to investigate their relationship with $CO_2$ emissions.

## 4. Results and Discussion

### 4.1. Comparative Analysis of Variables and Models

Figure 3 illustrates the detailed information of a different dataset at the same position. Compared with NTL, NUI-CV can enrich the detailed information within the city. To compare the reliability of $CO_2$ emissions, we employed a linear regression model to validate the estimated $CO_2$ from NTL and NUI-CV, respectively. Table 3 showed that the average $R^2$ of NUI-CV (0.74) was significantly higher than that of NTL (0.68). As a conclusion, NUI-CV is more suitable than NTL for estimating $CO_2$ emissions.

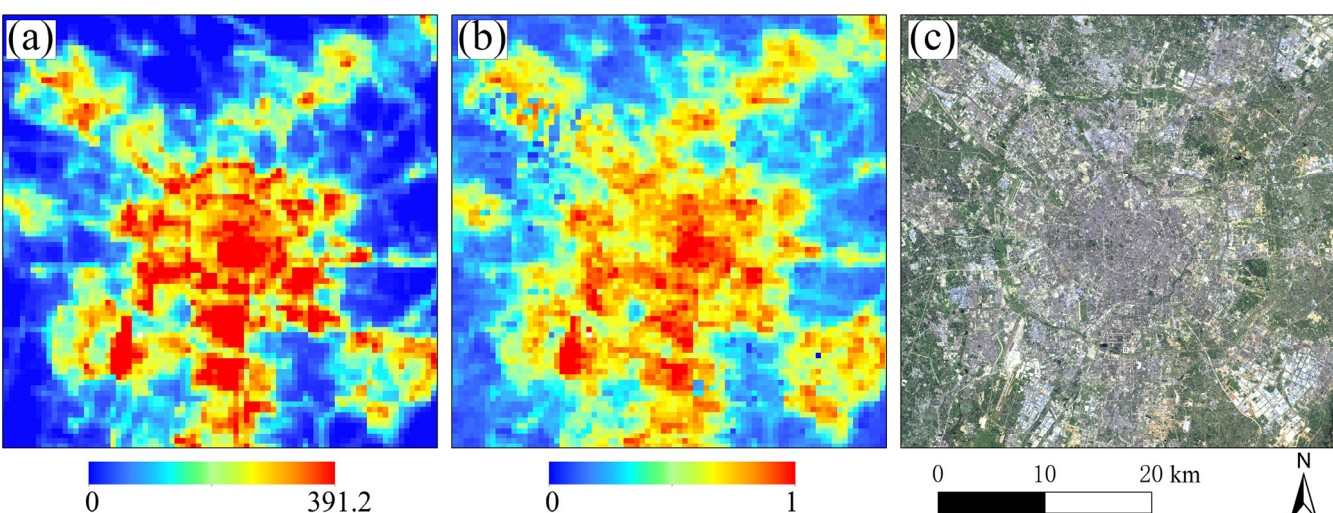

**Figure 3.** Comparison of three dataset in Chengdu City; (**a**) NTL; (**b**) NUI-CV; (**c**) Landsat 8 OLI image with 30 m spatial resolution.

**Table 3.** $R^2$ comparison of experimental results.

| Year | NTL | NUI-CV | Year | NTL | NUI-CV |
|------|------|--------|------|------|--------|
| 2000 | 0.5794 | 0.6461 | 2011 | 0.725 | 0.7718 |
| 2001 | 0.5336 | 0.6312 | 2012 | 0.6069 | 0.7035 |
| 2002 | 0.7525 | 0.7547 | 2013 | 0.7519 | 0.8251 |
| 2003 | 0.7706 | 0.7772 | 2014 | 0.7436 | 0.7907 |
| 2004 | 0.7643 | 0.7848 | 2015 | 0.6644 | 0.7716 |
| 2005 | 0.7096 | 0.7373 | 2016 | 0.6698 | 0.7554 |
| 2006 | 0.6908 | 0.7429 | 2017 | 0.679 | 0.7719 |
| 2007 | 0.763 | 0.7887 | 2018 | 0.6328 | 0.7259 |
| 2008 | 0.7148 | 0.7481 | 2019 | 0.616 | 0.7117 |
| 2009 | 0.5992 | 0.7075 | 2020 | 0.5829 | 0.6754 |
| 2010 | 0.7211 | 0.7666 | Average | 0.6796 | 0.7423 |

*4.2. Spatiotemporal CO$_2$ Emissions Dynamics*

4.2.1. Variations at National and Provincial Scales

The trend in CO$_2$ emissions in the majority of provinces is similar to the national trend, i.e., upward (Figure 4). However, some provinces (such as Beijing, Chongqing, Henan and Shanghai) have experienced a negative growth within recent years.

There are two possible reasons for this phenomenon: first, some provinces have undergone an industrial structural transformation, which has reduced CO$_2$ emissions; second, some developed provinces have reduced the local CO$_2$ emissions by moving industries with high CO$_2$ emissions to other provinces [12]. At the same time, some provinces (such as Shandong, Shanxi, Hebei and Inner Mongolia) have consistently high CO$_2$ emissions. This is due to the developed heavy industries, high-energy consumption, and low-energy efficiency in these provinces. Among these provinces, Shandong has a well-developed heavy industry, Hebei is dominated by the steel industry, and Shanxi and Inner Mongolia produce large amounts of coal [71]. These traditional industrial provinces with strong secondary industries inevitably produce large amounts of CO$_2$ emissions. It is difficult to change the industrial structure of a city, so the CO$_2$ emissions generated by energy consumption will not decrease quickly. However, this also shows us the way: optimizing industrial structure and increasing energy efficiency is the necessary method for traditional industrial provinces to achieve emission reduction goals.

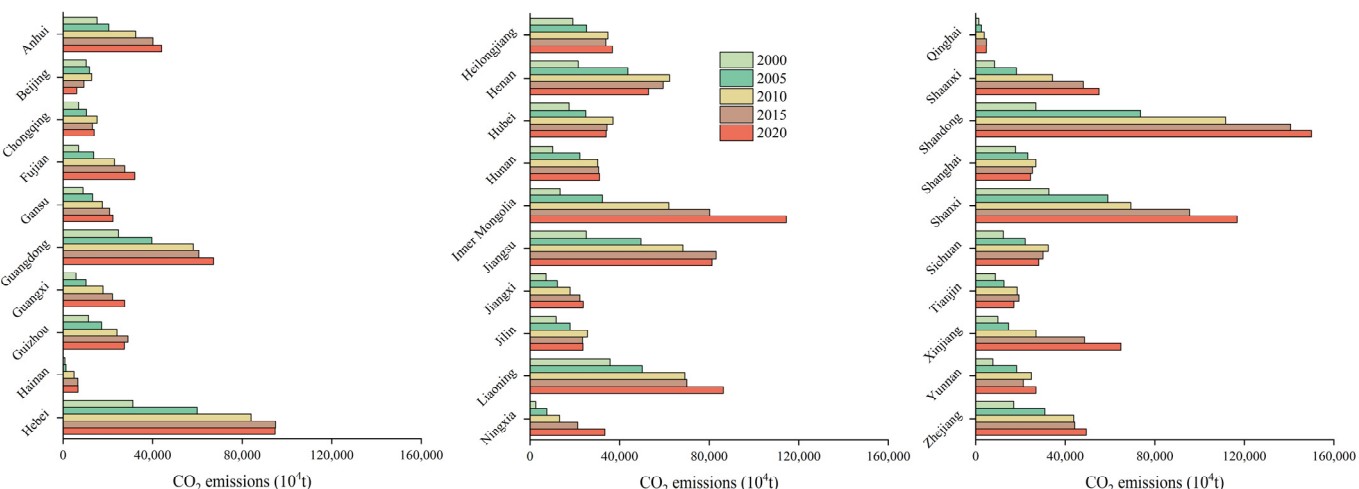

**Figure 4.** Statistical $CO_2$ emissions by province in China.

As illustrated in Figure 5, the spatial distribution of $CO_2$ emissions shows a significant change from 2000 to 2020. The areas of $CO_2$ emissions are mainly concentrated along the eastern coastal region and are relatively sparse in the western regions. The reason for this phenomenon is that the coastal areas are economically developed, with high population density and greater urbanization, resulting in intensive $CO_2$ emissions. As time goes on, there is an obvious expansion of $CO_2$ emissions and a high distribution in core urban areas and a low distribution in peripheral areas. This indicates that urbanization is increasing and that human activity is becoming more intense in core urban areas and is gradually expanding to the suburbs.

At the same time, it is shown in Figure 5 that the growth of statistical $CO_2$ emissions in 2020 is not significant compared to 2019. This is due to the COVID-19 pandemic in 2020 and the global economic slowdown, which has reduced the consumption of fossil fuels. However, such reductions have only slowed the growth rate of $CO_2$ emissions, rather than leading to a decline in statistical $CO_2$ emissions.

### 4.2.2. Study of Variation Trends at Pixel Scale

Although $CO_2$ emissions increased a lot in 2020 compared to 2000, in some urban core areas, $CO_2$ emissions showed a significant downward trend (Figure 6). Due to the increase in urbanization, the urban core cannot meet a city's development needs and, over time, some developed cities have entered the end of their industrialization development. Many people and industries are moving to the suburbs and the urbanization of the countryside is accelerating. Based on this shift, the industrial $CO_2$ emissions within the core urban areas are gradually decreasing, resulting in a reduction in emissions intensity. Due to the influx of a large number of people and industries, the suburban areas inevitably consume a lot of resources, resulting in a significant increase in $CO_2$ emissions.

Figure 7 shows the proportion of different categories of $CO_2$ emissions trends. Within the total study area, the proportion of extremely significant increase is the highest (0.467%), and the proportion of significant decrease is the lowest (0.008%). The categories of significant, slightly significant and not significant increase accounted for 0.133%, 0.034% and 0.105% in the study area, respectively. For the category with a decreasing trend, the highest proportion is extremely significant decrease, accounting for 0.054%. The proportion of slightly significant decrease and not significant decrease were 0.016% and 0.028%, respectively.

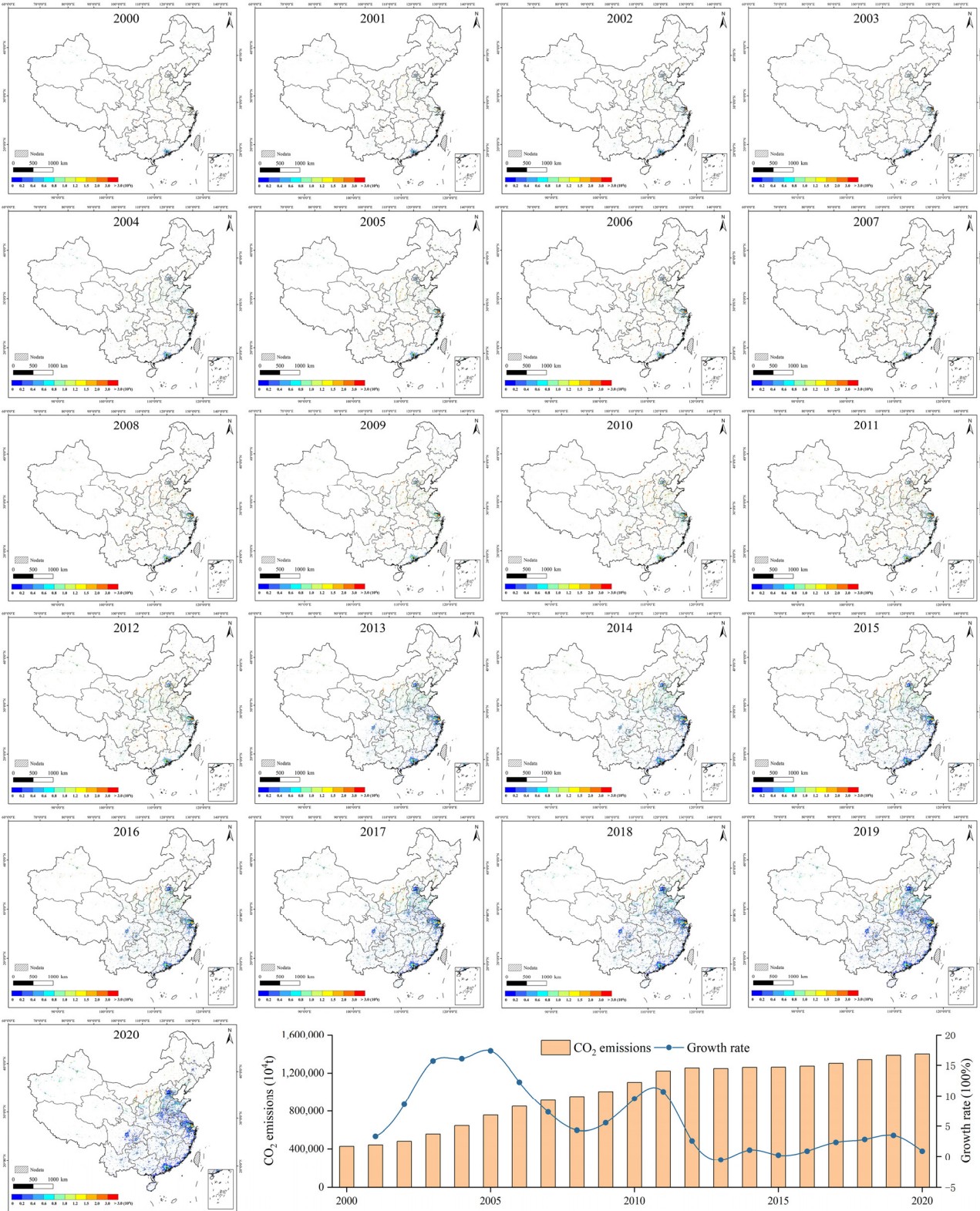

**Figure 5.** The estimated $CO_2$ emissions from 2000 to 2020 in China.

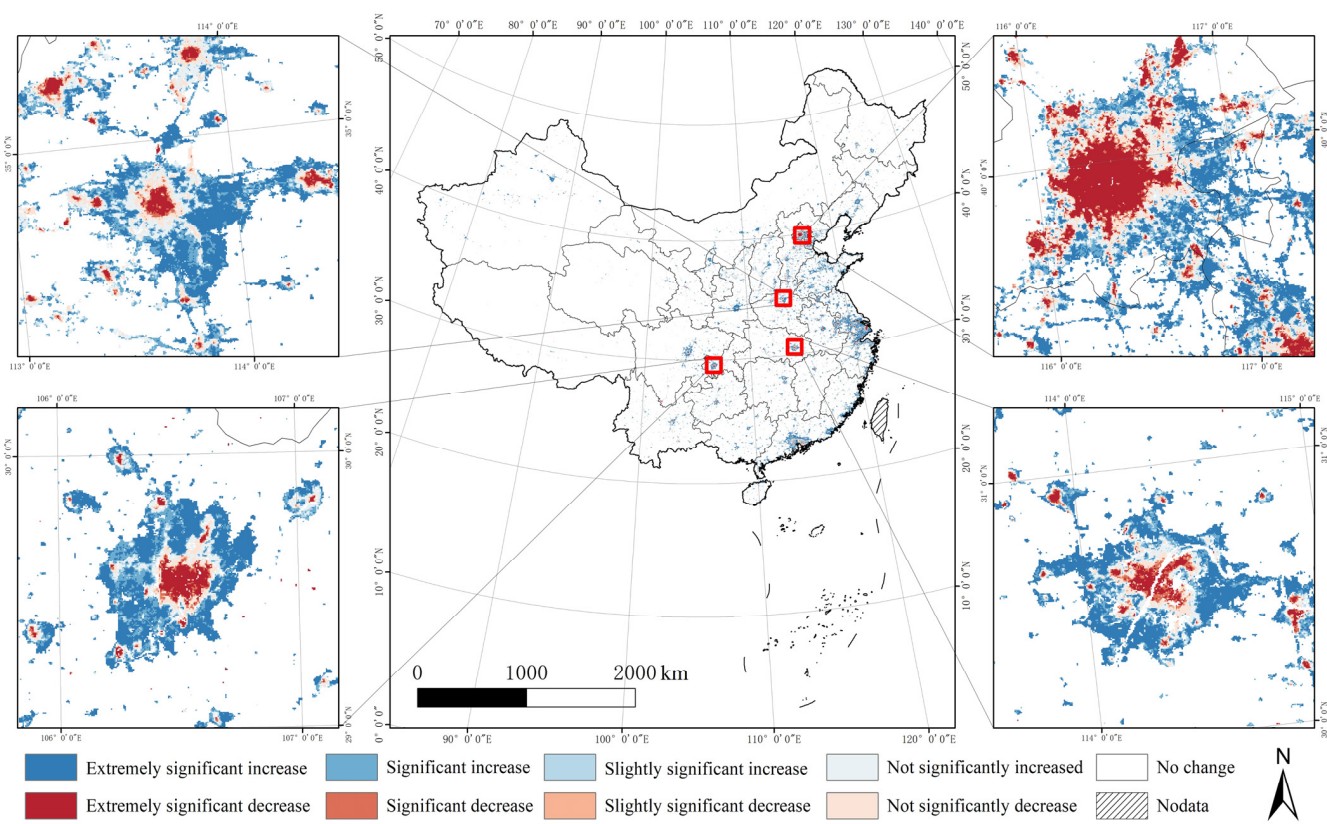

**Figure 6.** The spatiotemporal variations trend of $CO_2$ emissions from 2000 to 2020.

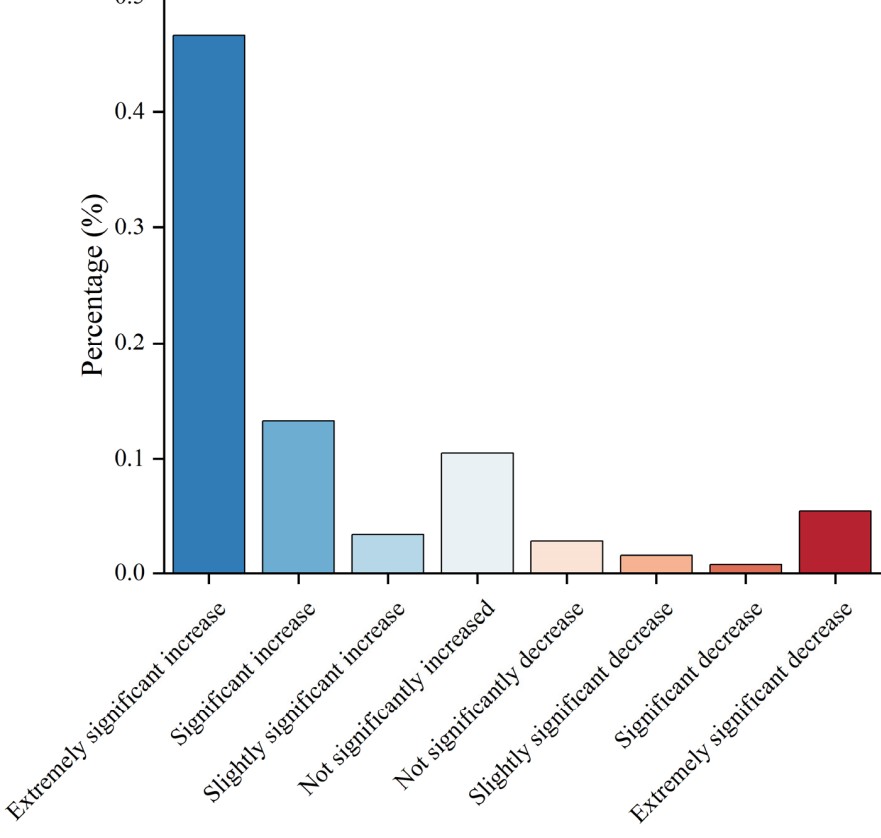

**Figure 7.** Percentage of different categories of $CO_2$ emissions trends.

### 4.2.3. Evaluation of The SDE Results

In Figure 8, the SDEs almost cover the central and eastern regions in China where $CO_2$ emissions are concentrated. The range of $CO_2$ emissions in 2020 is higher than that in 2000. Furthermore, the spatial distribution of $CO_2$ emissions shows a pattern of northeast–southwest directional polarization, and the directional trend of $CO_2$ emissions in 2000 is more obvious than in 2020.

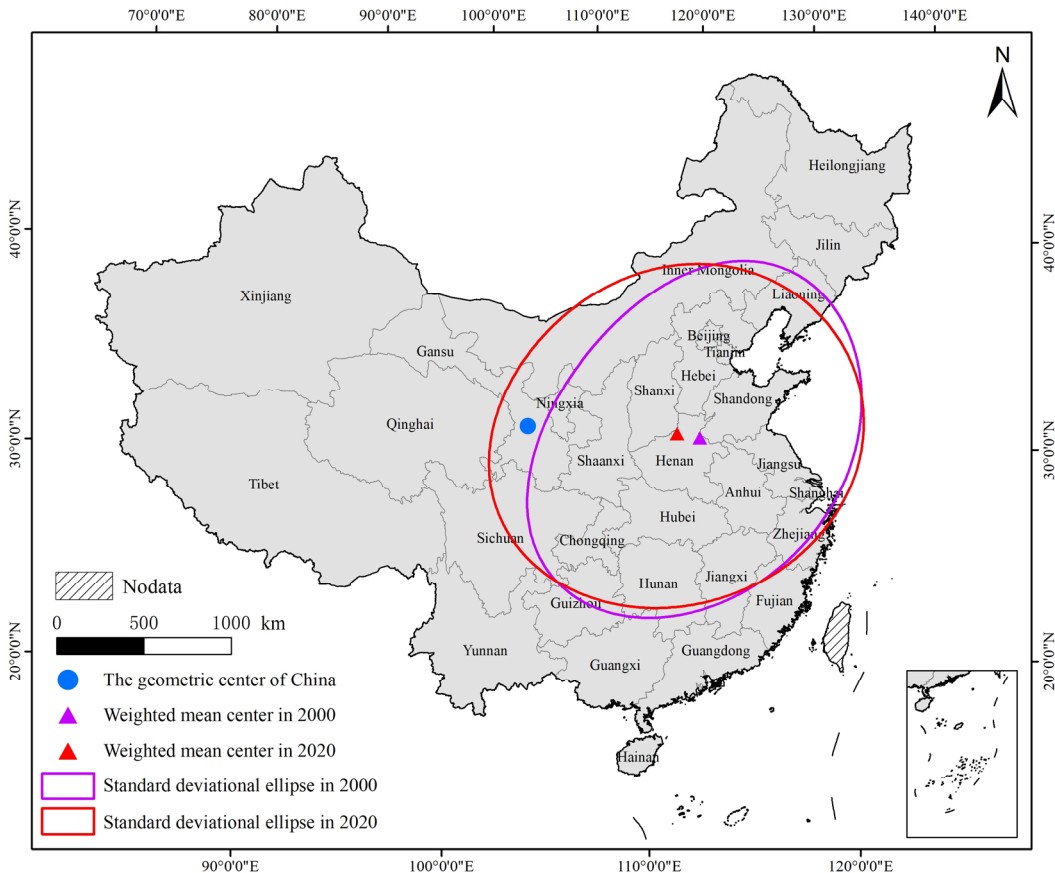

**Figure 8.** The SDEs at provincial scale in China between 2000 and 2020.

We also found that the weighted mean center of China's $CO_2$ emissions is located near Henan and Shanxi, rather than in the geometric center of China's land territory, Lanzhou. Lanzhou, located in Gansu Province, is not only the geometric center of China's land territory, but is also a key city in the western region [72]. This indicates that $CO_2$ emissions are imbalanced in spatial distribution, and the $CO_2$ emissions are higher in the east. In terms of the movement path of the weighted mean center, the center of national $CO_2$ emissions is moving to the west. This change may be due to China's "Western Development" and industrial transfer strategy. Due to its geographical location, western China is developing its economy at the expense of polluting the environment. The movement of some industries from the eastern coastal provinces to the west has enhanced local economic development, but has also generated significant $CO_2$ emissions. As a result, the national center of $CO_2$ emissions is moving to the west, and the directional trend of $CO_2$ emissions is weakening.

Although some scholars have simply analyzed the change of $CO_2$ emissions, significant analysis of trends in changes is still lacking. At the same time, few scholars have studied the migration of the weighted mean center of $CO_2$ emissions. The results of our innovative work have filled a gap in this field.

*4.3. Driving Forces of CO$_2$ Emissions*

To explore the drivers for CO$_2$ emissions, this study employed a spatial econometric model. The results for Log likelihood, Akaike info Criterion (AIC), and Schwarz Criterion (SC) are shown in Table 4.

**Table 4.** Result of spatial economics model.

| Variables | SLM | | | SEM | | |
|---|---|---|---|---|---|---|
| | **2000** | **2010** | **2020** | **2000** | **2010** | **2020** |
| Population | 0.1335 | 0.1838 | 0.6918 ** | 0.2162 | 0.2395 | 0.6985 ** |
| AVSI | 1.2972 *** | 1.1154 *** | 0.9205 ** | 0.8983 *** | 0.9851 *** | 0.9275 *** |
| AVTI | −0.6465 * | −0.579 ** | −0.9178 *** | −0.1862 | −0.4202 * | −0.8458 *** |
| $R^2$ | 0.8199 | 0.8141 | 0.6258 | 0.8472 | 0.8612 | 0.7072 |
| Log likelihood | −21.4209 | −18.1811 | −27.5137 | −19.7957 | −14.758 | −24.8366 |
| AIC | 52.8418 | 46.3623 | 65.0273 | 47.5915 | 37.5161 | 57.6733 |
| SC | 60.0117 | 53.5322 | 72.1973 | 53.3274 | 43.252 | 63.4093 |

Note: Significant at * 10% level, ** 5% level, *** 1% level.

AVSI significantly correlates with CO$_2$ emissions (Table 4), which aligns with the "high-energy consumption and high emissions" characteristics of the secondary industry. At present, the secondary sector is the main industry in China, and this will be difficult to change. This means that it will be difficult to see a significant reduction in CO$_2$ emissions in a short period of time because of the industrial structure. Although clean energy is being used and the industrial structure has improved in this regard, the impact of the secondary industry on CO$_2$ emissions is still high. Therefore, the industrial sector should continue to improve its industrial structure, in order to increase energy efficiency.

Only the population coefficient for 2020 passes the significance test at the 5% level. The results of SLM indicate that the population impact on CO$_2$ emissions is increasing year by year in 2000, 2010 and 2020 with coefficients of 0.13, 0.18 and 0.69, respectively. Population growth leads to more demand for food, housing, and transportation. These require more energy to meet the demands of industry, electricity, and transportation, resulting in more CO$_2$ emissions. These findings are also supported by other studies, which showed that the effect of population on CO$_2$ emissions cannot be ignored [73,74]. For some developed provinces, population size could be controlled as an effective way to control CO$_2$ emissions.

It is remarkable that the effect of AVTI on CO$_2$ emissions is always negative. This means that a strong development in the tertiary sector will help to cut CO$_2$ emissions. The financial and service industries are representative of the tertiary industry. These industries require less energy and produce less CO$_2$. This means that by keeping other variables constant, promoting the tertiary industries will help reduce CO$_2$ emissions. Therefore, promoting the transformation of traditional industries and supporting the development of the service and financial sectors should be the focus of government attention.

## 5. Conclusions

In this study, we examined the spatiotemporal dynamics of CO$_2$ emissions in China, and identified the regional heterogeneity of CO$_2$ emissions. Based on nighttime light images and MODIS NDVI data, the CO$_2$ emissions from 2000 to 2020 were estimated for the first time at 500 m spatial resolution in China. The model outputs showed that the proposed NUI-CV is more suitable for measuring CO$_2$ emissions than the traditional model (NTL). In addition, we evaluated the spatiotemporal dynamics and drivers of CO$_2$ emissions using the Theil–Sen and Mann–Kendall trend analysis, standard deviational ellipse and spatial econometric model.

Nationwide, China's CO$_2$ emissions are distributed unevenly, with more intensive emissions in the east. This phenomenon may be related to differences in regional development. The eastern region is economically developed and consumes a large amount of fossil fuels, leading to significant CO$_2$ emissions. The growth rate of CO$_2$ emissions in 2020

shows a significant reduction due to the impact of COVID-19 pandemic. Although the total amount of $CO_2$ emissions continues to increase, the $CO_2$ emissions in urban core areas do not rise, but show a downward trend. It is inspiring evidence for the achievement of carbon reduction targets. We also found that the national center of $CO_2$ emissions is moving to the west, and the directional trend of $CO_2$ emissions is weakening. Meanwhile, AVSI and population are positively correlated with $CO_2$ emissions, while AVTI has a negative correlation with $CO_2$ emissions.

## 6. Policy Implications

Our research provided some inspiration for carbon emissions reduction. According to the dynamic changes of $CO_2$ emissions in different cities, different emission reduction measures should be formulated separately. Energy efficiency and population size should become the focus of the government. Our results confirmed that the development of the tertiary sector is the key to reducing $CO_2$ emissions, and thus the relevant sectors should pay attention to this. Relevant departments should formulate a series of measures to promote the transformation of traditional industries and support the development of the service and financial sectors.

In conclusion, the impact of population and secondary sector on $CO_2$ emissions cannot be ignored. Raising residents' awareness of low-carbon approaches is also a crucial part of the reduction process. For underdeveloped provinces, economic growth and urbanization are the themes of development. Local governments should develop a series of policies that seek to protect the environment and develop the economy at the same time.

## 7. Limitations and Future Recommendations

However, there are still aspects of the study that can be improved. The first is the saturated image element problem of DMSP-OLS data, which limits their application. Although the nighttime lighting data used in this paper can solve this problem to some extent, the correction of DMSP-OLS data is the focus of upcoming research. Second, this paper only considers the population and industrial structure, and does not consider the effects of trade, policy, and capital flows on $CO_2$ emissions. Thus, the impact mechanisms behind $CO_2$ emissions still need to be further explored. We encourage scholars to study similar works of NUI-CV in other countries and regions.

**Author Contributions:** Conceptualization, Y.L. and W.G.; methodology, Y.L.; software, Y.L.; validation, Y.L., X.Z. and W.G.; formal analysis, Y.L., P.L. and W.G.; investigation, J.L.; resources, X.Z. and P.L.; data curation, Y.L.; writing—original draft preparation, Y.L.; writing—review and editing, W.G., P.L. and X.Z.; visualization, Y.L.; supervision, W.G.; project administration, X.Z. and W.G.; funding acquisition, W.G. All authors have read and agreed to the published version of the manuscript.

**Funding:** This research was funded by the National Natural Science Foundation of China (No. 41930650); State Key Laboratory of Geo-Information Engineering and Key Laboratory of Surveying and Mapping Science and Geospatial Information Technology of MNR, CASM (2021-03-04); The Ningxia Hui Autonomous Region Key Research and Development Project (2022BEG03064).

**Institutional Review Board Statement:** Not applicable.

**Informed Consent Statement:** Not applicable.

**Data Availability Statement:** Not applicable.

**Conflicts of Interest:** The authors declare no conflict of interest.

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
