# Peer review of "Exploring the Spatiotemporal Dynamics of CO2 Emissions through a Combination of Nighttime Light and MODIS NDVI Data"

_sustainability, doi:10.3390/su151713143_

Round 1

Reviewer 1 Report

The subject matter of the study is interesting due to the very topical issues concerning both the study of greenhouse gas emissions and the methods of analyzing such phenomena.

The research covers a very large area and uses an interesting methodological approach. In terms of content (study design, methodology, presentation of results and discussion), the manuscript is very well structured. Only the introduction is too limited to Chinese topics, which can be seen both in its content and in the list of references. The issues raised are of a global nature and this should be the nature of the first part of the introduction - later one can and should move on to topics related to the area of research.

It seems to me that the results necessarily require a critical discussion with reference to a number of other data on CO2 emissions in China, but also in other countries around the world.
In particular, the results published by the authors indicate a gradual increase in CO2 emissions in the period under consideration. However (for example, Figure 5), this increase is visible even for 2020 compared to 2019. 2020 (the first year of the COVID pandemic) is a period of significant economic slowdown, limited mobility of the population, disappearance of private and public transport, limiting the work of industry, reducing the consumption of fossil fuels. The authors should therefore logically and rationally explain the increase in CO2 emissions in China during this period observed in the study. A serious discussion in this regard is necessary, because the methodology used may have some weaknesses in terms of the reliability of CO2 emissions, but this is only my guess - I may be wrong. Perhaps the results and methodology are correct, and the results obtained are all the more valuable. The suggested brief discussion on this should clear up my (perhaps unwarranted) doubts.
I consider the lack of critical analysis of the obtained results to be the weakest point of the manuscript, which needs improvement.

A little note: Figure 5. "The estimated CO2 emissions in 2000 and 2020 in China". The current content of the caption indicates that the content of the figure applies only to two dates: 2000 and 2020. This is not true, we have separate figures for each year. Caption needs to be corrected.

Reviewer 2 Report

In the manuscript of “Exploring spatiotemporal dynamics of CO2 emissions through a combination of nighttime light and MODIS NDVI data” (Manuscript ID: sustainability-2585026), the authors estimated CO2 emissions by a normalized urban index based on combination variables (NUI-CV), which is based on the nighttime light and the Moderate Resolution Imaging Spectroradiometer (MODIS)normalized difference vegetation index (NDVI), and investigated the spatial and temporal dynamics and influencing factors of CO2 emissions over the period of 2000–2020 using Theil-Sen + Mann-Kendall trend analysis, standard deviational ellipse, and a spatial economics model The work presented is relevant to the Journal's field. The manuscript has got some potential. I would like to congratulate the authors for a considerable amount of work that they have done. Especially, the authors reported that NUI-CV is more suitable than NTL for estimating the CO2 emissions, with a 6% increase in average R2. The author also reported that the center of China's CO2 emissions lies in the eastern regions and is gradually moving to the west. In addition, the authors also uncovered that Changes in industrial structure can strongly influence changes in CO2 emissions, the tertiary sector playing an important role in carbon reduction. This manuscript has provided a new case to a better understanding of spatiotemporal dynamics of CO2 emissions through a combination of nighttime light and MODIS NDVI data. However, the manuscript needs further improved before to be accepted for publication. The reviewer has listed some specific comments that might be helpful of the authors to further enhance the quality of the manuscript. Please consider the particular comments listed below:

Comment 1, The authors need to improve the abstract. Therefore, the abstract should answer these questions about your manuscript: What was done? Why did you do it? What did you find? Why are these findings useful and important? Answering these questions lets readers know the most important points about your study and helps them decide whether they want to read the rest of the paper. Make sure you follow the proper journal manuscript formatting guidelines when preparing your abstract.

Comment 2, sections of Introduction. Although the section is well-structured and well-organized, the novelty of this paper should be further justified by highlighting main contributions to the existing introduction and literature review. This could be clearly presented in your related work. Please consider citing following papers entitled “Does urbanization redefine the environmental Kuznets curve? An empirical analysis of 134 Countries”; and entitled “Free trade and carbon emission revisited: The asymmetric impacts of trade diversification and trade openness”; and entitled “Linking trade openness to load capacity factor: The threshold effects of natural resource rent and corruption control”; and entitled “Digital economy and carbon dioxide emissions: Examining the role of threshold variables”; and entitled “Does renewable energy reduce ecological footprint at the expense of economic growth? An empirical analysis of 120 countries”; and entitled “The effects of trade openness on decoupling carbon emissions from economic growth–Evidence from 182 countries”, and paper entitled “Trade openness helps move towards carbon neutrality—Insight from 114 countries”. There has already been a large number of literatures related to your research, i.e, Bibliometrics analysis. Therefore, it should be better elaborate the contribution of the work to the existing literature, so as to further bridge the gaps between the research background and research purposes.

Comment 3. The research methodology could benefit from further improvement. It is important to provide a clear justification for the methodology approach used, explaining why it was chosen and how it is appropriate for the research question at hand. Additionally, it would be helpful to reference prior studies that have successfully used this methodology approach to strengthen the argument for its use in this particular study.

--The data section requires improvement. The authors must address several key questions to provide a better understanding of their approach. Specifically, why were these variables selected for the model? What does the existing literature say about these variables? Additionally, it's important to provide information on previous authors who have used these variables. Without this information, readers may find it difficult to fully comprehend the approach and results presented in the study.

Comment 4. section of Results and discussion. The section is well-structured and well-organized. However, it would be better to discuss what your findings are different from the past works. A comparison with the results of the previous paper would further enhance the innovative nature of the paper.

Comment 5. The authors need to improve the quality of the conclusions section. The conclusions section needs to be supported by the results and the authors need to show how their investigation advances the field from the present state of knowledge.

- To provide more comprehensive and actionable recommendations, the authors should create a dedicated subsection titled "Policy Implications." In this section, they should identify the specific areas where the current policy falls short and explain why their proposed recommendations can help improve the status quo. It's important to keep in mind that policymakers are interested in practical, cost-effective, and socially acceptable solutions, so the authors should address the following questions before presenting their recommendations:

What specific changes need to be made?

How will these changes be implemented?

What resources will be required to implement the changes, and where will they come from?

What are the overall benefits of the proposed changes for policymakers and society as a whole? By answering these questions, the authors can provide a more compelling and practical set of policy recommendations that can help address the shortcomings of the current policy and lead to positive changes.

-  The authors should consider creating a new subsection titled "Limitations and Future

Recommendations". It's essential to address the study's limitations, which are the design or methodology constraints that may have affected the interpretation of the research findings.

Limitations may have an impact on the ability to generalize results or describe applications for practice, as well as the usefulness of the findings that resulted from the research design or method used to establish internal and external validity, or unanticipated challenges encountered during the study. In addition to addressing limitations, future recommendations should consider the following aspects: (1) building upon a specific finding in the research; (2) addressing a flaw in the research design; (3) testing a theory, framework, or model in a new context, location, or culture; (4) re-evaluating or (5) expanding a theory, framework, or model. It's important to consider these aspects to ensure that future research is based on solid foundations and provides valuable insights that can inform practice and policy decisions.

- The authors should prioritize improving the presentation quality of the manuscript,

Comment 6. particularly with regards to the organization of the text and the presentation of tables and figures. While the manuscript shows promise, its overall investigation quality requires improvement. The authors must ensure that the manuscript is both attractive and readable, in order to increase its likelihood of being read and cited. Paying close attention to details in all manuscripts will be critical to achieving this goal.

Comment 7, There are still some occasional grammar errors through the revised manuscript especially the article ''the'', ''a'' and ''an'' is missing in many places, please make a spellchecking in addition to these minor issues. In addition, some sentences are too long to be easy to read. It is recommended to change to short sentences, which are easier to read.

Comment 8, References. Please check the references in the text and the list; You should update the reference. Please read the latest published papers carefully and format your references according to the format required by Sustainability. If this revised paper is sent to me for re-review, the first thing I will check the references.

Moderate editing of English language required

Round 2

Reviewer 1 Report

I have read the responses to the comments in the review, as well as the revised version of the article. The new version of the manuscript has been significantly improved as a result of taking into account the comments contained in individual reviews. The current version is in my opinion suitable for publication.

Reviewer 2 Report

The authors have incorporated comments from the first round of review. My concerns from my previous review have been addressed. I would recommend the paper to be accepted for publication.  

Thank you,

Best regards,